# Antibiotic Stewardship in Surgical Departments

**DOI:** 10.3390/antibiotics13040329

**Published:** 2024-04-04

**Authors:** Irene Kourbeti, Aikaterini Kamiliou, Michael Samarkos

**Affiliations:** Department of Internal Medicine, School of Medicine, National and Kapodistrian, University of Athens, 11527 Athens, Greece; aikkamiliou@gmail.com (A.K.); msamarkos@gmail.com (M.S.)

**Keywords:** antimicrobial resistance, antimicrobial stewardship, surgical site infections, surgical antibiotic prophylaxis, diagnostic stewardship, biomarkers

## Abstract

Antimicrobial resistance (AMR) has emerged as one of the leading public health threats of the 21st century. New evidence underscores its significance in patients’ morbidity and mortality, length of stay, as well as healthcare costs. Globally, the factors that contribute to antimicrobial resistance include social and economic determinants, healthcare governance, and environmental interactions with impact on humans, plants, and animals. Antimicrobial stewardship (AS) programs have historically overlooked surgical teams as they considered them more difficult to engage. This review aims to summarize the evolution and significance of AS in surgical wards, including the surgical intensive care unit (SICU) and the role of diagnostic stewardship (DS). The contribution of AS team members is presented. The new diagnostic modalities and the new technologies including artificial intelligence (AI) are also reviewed.

## 1. Introduction

Antimicrobial agents are powerful weapons used to help us fight infections [1]. AMR is one of the leading causes of death around the globe. Improving patient safety in health care aligns with the approach used to combat this phenomenon [2]. Infection-related mortality is expected to exceed 10 million cases per year by 2050 [3,4]. This is a huge crisis that society is facing, and it is caused largely by the unabated overuse of antimicrobial agents [4,5,6]. The significance of this crisis led the United Nations to adopt resolutions that commit to national action plans (NAPs) for AMR [7]. NAPs should be guided by country-level needs and the existing infrastructure [7]. The end point should always be the optimization of antibiotic use [6,8].

There are three points when selective pressure can be reduced: before therapy, where we should only treat the patients who are truly infected; during therapy, where we should avoid the use of combinations when only one agent suffices; finally, at the end of the therapy, by treating the patient for as long as required [1]. The postantibiotic era of antimicrobial resistance poses the threat death after even a minor surgery [1,9]. Routine operations like hip replacements or complicated procedures such as organ transplants could be deadly because of the infection risk [1,10]. Up to 50% of the antibiotics used in the hospital environment may be unnecessary [11,12]. This is mostly due to the extension of the recommended durations of treatment or to treatments that are targeted toward colonizing or contaminating pathogens [11]. Prolonged and inappropriate antibiotic courses appear to be the key factor in the rise in antimicrobial resistance across the globe [13]. Infection control and prevention in the hospital setting have been recognized as among the key patient safety and public health quality indicators [14]. Although there has been significant progress in the organizational level, hospitals need to establish AS programs that are characterized by the necessary sustainability and resilience [14]. This can be achieved by ensuring the widest participation in and commitment to the program by a diversity of specialists that include infectious disease (ID)-trained physicians and pharmacists, nurses, surgeons, intensivists, and hospital administrators [2,8,12]. Although ID physicians and pharmacists traditionally provide advice on infection management, the role of the other members cannot and should not be overlooked [2].

This review aims to present the special challenges faced by AS teams in the departments, as well as updated data on rational antibiotic use and shorter treatment durations for surgical infections. Reference is also made on the special roles of each member of AS teams and the importance of diagnostic laboratories.

## 2. Results and Discussion

### 2.1. Antimicrobial Stewardship Principles and Practices

Healthcare-associated infections (HAIs) affect around 5–15% of the patients admitted to hospitals [15]. These patients are extremely vulnerable to acquiring multidrug-resistant organisms (MDROs), which may lead to untreatable infections [15]. HAIs are responsible for more deaths worldwide than malaria, tuberculosis, and AIDS combined. Their burden is higher than that of 32 major communicable diseases, and the total annual global cost is estimated as around USD 10 billion [16]. Infection prevention does not constitute a necessity only for high-income countries: it should also be implemented in lower- and middle-oncome countries (LMICs), where hospital-associated resistant infections are equally relevant and account for a higher percentage among the HAIs; however, it is not feasible to translate policies from high-income countries (HICs) to LMICs [15,17,18].

The predefined objectives of an AS program include the following: 1. suggestion for empirical therapy according to the guidelines; 2. de-escalation of therapy based on cultures; 3. adjustment of therapy according to renal function; 4. switching from intravenous to oral therapy; 5. therapeutic drug monitoring; 6. discontinuation of empirical treatment when there is no evidence of infection; 8. preparation of a local antibiotic guide that includes a list of restricted antibiotics; and 9. bedside consultation [12]. The end point should be the achievement of optimal clinical outcomes related to antimicrobial use and limiting the selective pressure on bacterial populations that drives the emergence of antimicrobial-resistant strains [19]. The modern vision of an integrated stewardship model demands a multidisciplinary approach [20,21,22]. The common aim of the AS team and everybody else involved in the process is responsible antibiotic use [23].

The early detection and close surveillance of MDROs and the reaction to every possible transmission are very important tasks [21] that depend on prompt access to rapid microbiological diagnostics and the collaboration between infection control and medical microbiology personnel. Together, they are responsible for the implementation of hand hygiene, the detection of outbreaks of resistant pathogens, the isolation of patients with MDROs, and the investment in hospital-wide infrastructure [8,15,16,20]

Evidence-based guidelines should be developed by each institution and implemented for the purpose of reducing infections and AR [15]. Education programs should aim to train personnel on the guidelines and supervise their implementation [15]. A key challenge for infection prevention professionals is to implement scientific evidence in daily practice because scientific evidence is not sufficient to promote change [15]. What we usually observe is an implementation gap between the literature and clinical practice. Organizations tend to prefer the status quo, and AS teams follow procedures that affect a range of employees in various professions and levels [15], thereby dictating the usage of subtle tactics and continuous reminders that practice recommendations are evidence-based [15].

AS should be endorsed by leaders and hospital administration. In order to generate institutional support, AS members should present a business plan, identify barriers, and provide strategies for resolution. They should be able to document improvements in quality of care using metrics [8]. The common aspect of successful AS interventions is tailoring to local conditions [24]. Overall, AS can safely reduce unnecessary antimicrobial use in the hospitals; however, this reduction should be implemented through behavior-changing techniques [25].

### 2.2. The Impact of Infections in Surgical Wards

AS initiatives in surgical wards should initially focus on surgical site infection (SSI) prevention and surgical antibiotic prophylaxis (SAP) [2,4]. SSIs are infections of the incision or organ or space, which occur in 1–3% of patients undergoing inpatient surgery, with the burden being worse in acute-care surgery [26,27]. They contribute the most to hospital care costs [28]. In LMICs, they may affect 8–30% of surgical patients, making them one of the most common HAIs [29,30]. The higher rates are mainly due to poor healthcare quality [17,31]. The challenges LMICs face include issues related to hand hygiene, alcohol rub shortages, and the lack of awareness among healthcare providers [31]. Overall, SSIs comprise 14–17% of all HAIs [29]. They are viewed as a major contributor to patient mortality, length of stay, and health care costs; up to 60% of them are considered preventable [4,26,29,32,33,34]. Nowadays, surgical patients present with complex comorbidities, and 77% of the deaths in patients with SSIs are directly attributed to them. The higher fatality rates in LMICs also reflect the unavailability of second-line drugs [30,32,33]. Since the human and financial costs of SSIs are increasing, the prevention of SSIs has been established as one of the areas of focus of quality improvement initiatives, and there is increasing demand for evidence-based interventions [9,26,29,32,35]. The main efforts should emphasize the mitigation of the risk of SSI in the AR setting [4].

Prevention efforts should target all surgical procedures, but they should mostly emphasize those that come with increased human and financial burdens [32]. For instance, an increasing number of elderly people are undergoing prosthetic joint arthroplasties. By 2030, it is expected that the infection risk of hip and knee arthroplasties will increase from 2.18% to 6.55 and 6.8%, respectively. Since the most common indication for prosthetic joint revisions is infection, the costs are expected to escalate [32].

An analysis of National Nosocomial Infections Surveillance (NNIS) system data revealed that the most common pathogen that causes SSIs is *S. aureus*. The increase in MRSA proportion has led to increased mortality rates and longer hospital stays [4,26]. *E. coli* and *Pseudomonas aeruginosa* are the next most common pathogens; infections caused by resistant *K. pneumoniae*, *P. mirabilis*, *E. aerogenes*, and no-fermenting Gram-negative pathogens regularly occur [4,36,37] (Table 1). The percentage of resistant pathogens among the main bacteria responsible for SSIs has varied over time [38] (Table 2). Surgical intervention and proper antibiotic utilization have significantly reduced mortality rates in people with intra-abdominal infections [39]. The reduced consumption of antibiotics, along with rapid discontinuation and de-escalation to the oral route, is also accompanied by significant cost savings [40,41].

One of the major aims of a team is a shift toward narrow-spectrum drugs and the decrease in the relative consumption of carbapenems, especially in the SICU setting. When carbapenems are administered empirically, every effort should be made to exclude extended-spectrum beta-lactamase (ESBL) infection and de-escalate sooner [42]. The development of reliable predictive scores for ESBL infection is imperative in order to further limit the use of carbapenems and new antibiotics. Efforts to provide more-focused treatment will translate eventually to a relevant change in prescribing habits [41]. The above achievements always come at a cost. But, the fear of withholding or stopping treatment for a patient who really needs it is an even greater challenge. The stewardship philosophy should never put patients at risk [41].

### 2.3. The Use of Antibiotics in Surgical Departments—The Importance of Education and Prescribing Etiquette

Guidelines exist that provide a robust framework to establish good clinical practices [2]. Despite the existing guidelines, there are several factors that hinder proper antibiotic use in surgery. Guidelines may fail to offer a clear indication for proper antibiotic use, and doctors do not always adhere to the guidelines, exhibiting defensive treatment practices. In addition, the cooperation between surgeons and the AS team may vary, and surgeons may fail to use laboratory resources and inflammatory biomarkers effectively [41,43,44,45]. As a result, antibiotic prescription in surgery lacks clarity and occurs in settings of disjointed information and low policy adherence [45]. Despite the multiple campaigns aimed at creating awareness of the consequences of overprescribing, antimicrobial consumption densities remain significant higher in comparison to the percentage of patients with infection in surgical ICUs [41]. In addition to the above, antibiotic prescribing can be a very complex process that is prone to errors and subjected to unwritten rules such as clinical autonomy and hierarchy [43,44,46]. The above challenges are considered very significant in the nosocomial setting and especially in surgical departments [1,47].

In the USA, it is estimated that a 30% reduction in the efficacy of antibiotic prophylaxis, in comparison to the effect sizes recorded between 1968 and 2011 for ten major surgical procedures, would result in 120,000 additional infections per year [9]. AS programs in surgery should be implemented starting with the need, choice, duration, and timing of prophylactic antibiotics [26,40]. The reason for the use of prophylaxis is the prevention of the infection [4,9,26]. In acute-care hospitals, about one-third of patients receive antibiotics. It has long been proposed that the simplest and safest way to reduce the antibiotic consumption is to use them for as long as necessary in order to achieve optimal cure rates [1,5]. The efforts of specialists and AS teams should emphasize convincing clinicians to not prescribe antibiotics for long periods. This is likely to be the most palatable method for practicing clinicians [1,5,7,26]. The end point is the reduction in selective pressure on the prevalent microbial flora [1]. The growing incidence of AMR increases the risk of SSIs complicated by resistant bacteria, the adverse outcomes owing to prolonged lengths of stay, the requirement for novel antibiotics, the rates of surgical revision and mortality, and the associated hospital costs [4,48].

Perioperative prophylaxis plays a critical role in the prevention of SSIs, but its use comes with the risk of adverse events, *C. difficile* infection, and antimicrobial resistance [49]. Prophylaxis prescriptions account for 9–15% of the total antibiotic use in the hospitals, but SAP is considered an area of variable practice [49,50,51]. Older studies identified variable compliance rates with the guidelines that varied from 25% to 80%. More recent studies in the USA reported an overall adherence rate of 59%, even when accounting for patient allergy and MRSA status, with a median hospital adherence rate at 64% [49]. The adherence rates decreased significantly between 2019 and 2020. The COVID-19 pandemic was independently associated with poor compliance to guidelines in some studies [49], whereas, in others, an improved agreement was noted as far as it concerns the choice and duration of prophylaxis, as well as the use of restricted antibiotics, including carbapenems [52].

There are criteria for the categorization of surgical procedures according to the possibility of the development of an infection. Wound contamination assessment is undertaken prior to and during surgery [4]. Antimicrobial prophylaxis is mainly indicated in procedures associated with a high rate of infection and in certain other procedures where the consequences of infection could be serious, even if unlikely (e.g., in the case of prosthetic implants) [35]. Cardiac surgery, placement of vascular prostheses, craniotomy, and total hip arthroplasty are examples of such procedures. Prophylaxis is absolutely indicated for clean-contaminated procedures. In dirty surgery, prophylaxis is considered as therapy, and a targeted treatment approach is generally indicated [4]. The ideal prophylaxis drug should be highly efficacious, low in toxicity, and inexpensive and have the narrowest spectrum of activity. The antimicrobial agent selected should be active against the pathogens most likely to cause infection and deliver adequate antibiotic exposure throughout the surgical site [4]. Individual institutions should consider the efficacy, safety, and the acquisition costs when implementing prophylaxis guidelines [35]. The safety profile and the patient’s allergies should always be considered.

For most procedures, cefazolin is the drug of choice [36,49]. The adoption of recommendations, however, should always depend on local epidemiology. The knowledge of local susceptibilities is important, especially the percentage of methicillin-resistant *Staphylococcus aureus* (MRSA) [50]. MRSA comprises around 20% of *S. aureus* isolates at the global level, but, in some regions, reports reveal percentages up to 80% [4]. Vancomycin has been the most commonly misused antibiotic agent, accounting for 77% of all nondherent surgical prophylaxis regimens. Most commonly, vancomycin is unnecessarily added to cefazolin [49]. Since 1999, the control and prevention of vancomycin resistance have been a major issue with advisory committees. Vancomycin is not recommended for routine use in any procedure. However, vancomycin may be considered in patients with known MRSA colonization or at high risk of MRSA colonization, especially if the surgery involves prosthetic material [4,33,35,49,53]. Its use should also be considered in settings of high MRSA prevalence or in proven MRSA outbreaks but not in a suspected MRSA outbreak [33,43,53,54]. When no MRSA is implicated, there is no difference in the SSI rates between when a glycopeptide or a β-lactam is used [33]. A fixed dose of vancomycin of 1 g has been associated with increased rates of SSIs in cardiothoracic and orthopedic surgery studies. Thus, a weight-based dose of 15 mg per kilogram of body weight is recommended [54]. The addition of vancomycin to cefazolin is not superior to placebo when used in arthroplasty without known MRSA colonization [54]. However, cohort studies suggested that infections with MRSA may be mainly due to hospital acquisition rather than community colonization [54]. Institutions should develop individual guidelines for proper vancomycin use. A thorough allergy history is necessary: it is problematic that in a recent U.K. study, in two-thirds of patients with documented penicillin allergy, there were no details recorded [4]. Patients who report a β-lactam allergy are at higher risk of SSIs and *C. difficile* infection due to the use of alternative broad-spectrum and often inferior antibiotics [4,33,55,56]. Among these patients, only a small proportion have a true allergy, and the proper tools should be utilized for the delabeling [55,56].

Optimal timing is considered key to SAP [4]. In a study on hip arthroplasty, timely administration was proven to be the most important factor in the quest for SSI prophylaxis [57]. The concentration at closure is equally vital for the avoidance of SSIs [58]. The latest recommendations favor the administration of a preoperative dose within 60 min of surgical incision. In the case of vancomycin and fluoroquinolone use, the administration should begin within 120 min, or, in patients undergoing arthroplasty, 45 min before the incision is appropriate [29,33,54]. The recommended doses may need to be amended for patients with obesity (a particularly high-risk group for SSI, which accounts for one in five surgical patients), and the dose should be repeated during procedures exceeding three hours in duration in order to ensure adequate serum and tissue concentrations [4,35]. The recommended dosing involves a single dose or continuation for less than 24 h. According to some authors, ideally, antimicrobial prophylaxis should be discontinued at the time of closure in the operating room [33,49,51]. The presence of a drain or the placement of a prosthetic device is not an indication for the continuation of antibiotic prophylaxis [34]. Unnecessary prolonged use offers no gain in efficacy, and it is associated with higher costs, increases in resistance, and increases in the associated risks such as *C. difficile* infection [4,50,59,60]. When prophylaxis is used the proper way, the risk of *C. difficile* infection is not higher [61]. Compliance with SAP policies has generally been lower in LMICs [62].

The European Surveillance of Antimicrobial Consumption protocol [ESAC] determined the duration of surgical prophylaxis as one of the indicators driving improvements in the quality of prescribing [50]. In a recent Scottish study, the duration of surgical prophylaxis was compliant with the rules in 68.6% of the cases, a much better percentage compared to the European one reported by the ESAC (48.1%) [50]. When they examined the compliance with the perioperative single dose, with an emphasis on colorectal surgery, Scottish hospitals performed better than European hospitals (49.3% vs. 27% compliance) [50]. Cardiothoracic surgery was the subspecialty with the greatest compliance rates in this study (95%). In a recent U.S. study, hip replacements had the highest compliance rate (65%), with spinal procedures presenting the lowest (49%) [49]. The use of standardized order sets, the automatic stop-order settings, as well as the educational initiatives have facilitated the adoption of guidelines for SAP [26]. Such modalities that also include checklists differ between LMICs and higher-income settings [30]. Agodi et al. published a comprehensive review of 28 studies on the compliance with the indication, timing, selection, and duration of perioperative prophylaxis. The summary of the studies included in this review is shown in Table 3 [63].

The need and duration of antibiotics administration have long been significant issues in surgical departments. Surgical intervention is important for achieving source control in intra-abdominal infections. This can be problematic in low-resource areas, where the availability of surgical services is limited [64]. In the study by Sawyer et al., it was proven that the outcomes in patients with intra-abdominal infections where source control was adequate and received a fixed short course (four days) of antibiotics did not differ in outcomes from those who received a more prolonged course of eight days [65,66]. This underlined the significance of source control, and it proved that the benefit of antibiotic administration is limited to the first few days after surgery [27,65]. If patients have signs of peritonitis after 5 to 7 days of antibiotic use, diagnostic investigation is warranted in order to address an ongoing uncontrolled source of infection [13]. Other potential targets of the AS in intra-abdominal infections (IAIs) include the limitation of unnecessary broad-spectrum antibiotics, the prompt transition from intravenous to oral therapy, and the recommendation to avoid specific antibiotics in settings with high resistance rates [66].

Patients with uncomplicated appendicitis (simple suppurative appendicitis without gangrene, perforation, or abscess) can be managed with antibiotic treatment alone [39]. The proposed duration of antibiotic treatment is 1–3 days of intravenous antibiotics, followed by 7–10 days of oral antibiotics [67].

The recommendation for antibiotic use in acute uncomplicated diverticulitis, defined as an episode of a short history without sepsis and no signs of an abscess, free air, or fistulas in the CT, has so far been based on tradition and expert opinions [68]. However, in a randomized trial, there were no significant differences in the frequency of perforation, abscess formation, or need for surgery between the groups that received and those who did not receive antibiotics [68]. In the same manner, antibiotics did not prevent complications, accelerate recovery, or prevent recurrences [68]. The authors concluded that antibiotics should be reserved for patients with complicated disease [68].

Antibiotic treatment in acute pancreatitis has been widely investigated but its misuse has evolved into a global challenge [13,69]. Current guidelines do not recommend the use of prophylactic antibiotics [70]. Taking into consideration local infectious complications along with extrapancreatic indications, antibiotic use could be justified in 20–40% of patients with acute pancreatitis. Patients with necrosis but no signs of organ failure and infection do not benefit from antibiotic administration, even if the evolution to severe or necrotizing pancreatitis is imminent [71]. However, up to 77% of the patients with severe pancreatitis may receive antibiotics, with 66% of them not even manifesting any signs of an infection [13,72].

Antibiotic stewardship is a very important issue in patients undergoing surgery for solid organ transplants (SOTs) [73,74]. The consequences of inappropriate antimicrobial use such as drug toxicities, MDR organism infection, and *C. difficile* infection (CDI) disproportionately affect recipients of SOT, and they are associated with poor allograft and patient outcomes [75]. Even though there are ongoing efforts by transplant societies to implement specialized principles, there is still a lack of guidelines for this population [74,76,77]. Prophylaxis in SOT should ideally be tailored to pre-existing known colonization or chronic infection. Cultures of the transport fluid may also help [73]. With appropriate interventions regarding surgical prophylaxis and feedback to the surgical departments, AS has proven beneficial in decreasing SSIs, the number and costs of antibiotics used, and antimicrobial resistance in SOT [76]. This has mainly been achieved through the discontinuation of unnecessary antibiotics [78]. Evidence is mounting for the use of SOT-specific antibiograms in guiding empirical choices, monitoring resistance patterns, and reducing the impact of differential antimicrobial susceptibilities on formulary choices [76]. Since the population undergoing SOT is more prone to surgical infections in the immediate postsurgery period, it is very important to use specific antibiograms and be aware of the differences in the susceptibilities of Gram-negative pathogens [76].

There are significant gaps in the literature on targeted decolonization strategies for multidrug-resistant Gram-negative pathogens (MDR-GNBs) in surgical patients. Thus, no routine decolonization of MDR-GNB carriers is recommended. More randomized control trials are needed, especially in high-risk populations such as those undergoing SOT [9,79,80]. Although there is a conditional recommendation for rectal screening and adjusted perioperative prophylaxis in patients undergoing colorectal surgery and SOT, there is limited guidance for other patients colonized with resistant bacteria [4,9,81]. The screening includes ESBL pathogens and CRE if indicated by the local epidemiology [80]. In clean surgeries, MRSA decolonization 5 days prior to surgery (using chlorhexidine body wash and nasal muripocin) decreased MRSA infection by 17% [4].

The culture around antibiotics is defined by superficiality in following orders or lapsing in patterns of prescription [82]. Clinicians perceive antibiotic resistance to be a national problem but less relevant to their own institution [82]. Staff education comprises a very important part of antimicrobial AMS programs, but it may also be a challenge in LMICs [30,39,82]. Education may include continuous training programs on AMR, prescription instructions for surgical prophylaxis and treatment, and the distribution of algorithms on empirical therapy [40,66]. When education is part of an AS implementation strategy, the adherence rates in surgical departments may approach 85% [48,66,83]. It is very important that the AS team first presents evidence on the efficacy and benefits of shorter antibiotic courses. They may face the frustration that there is no definite proof of the efficacy of shorter regimens; however, the proof on this subject is mounting, including studies in patients who are immunosuppressed [1,84]. In the long term, new evidence will alter prescribing practices, which will represent a cultural shift [1,8,84]. In a recent paper that examined the perceptions on AMR and AS of physicians in medical and surgical wards, the doctors’ attitudes were similar. These results demonstrated the impact of the correct approach in education and feedback, and they represent a shift from attitudes previously reported [85].

Research has highlighted the dynamics that govern the antibiotic dynamics in surgical wards [4]. Apparently, the knowledge of junior doctors on antibiotic prescribing and infection prevention is poor, and local resistance and antibiotic misuse are usually under-estimated [2,15,46,86]. Antibiotic decision making is perceived as a nonsurgical intervention that can be allocated to junior doctors or other specialties as the senior surgeons are usually in the theater [87]. This cultural etiquette imposes an autonomous position on senior doctors and a tendency for noninterference when an antimicrobial has been prescribed by a peer [44]. In addition, surgical patients are less likely to have their antibiotics reviewed during the rounds [7]. This individualistic and hierarchical culture and the issue of ownership of the antibiotic prescription may limit integrated care in surgical settings [4]. The clinical leaders that guide AS and education programs should be setting goals to help standardize and improve prescribing behaviors through collaboration among the healthcare workers, with patient safety being the strategic goal [2,4,13]. All the years of overprescribing antibiotics require a behavioral change. Simply asking clinicians to do a better job with antibiotics has not and does not work.

### 2.4. The Microbiology Laboratory and Diagnostic Stewardship—The Role of Biomarkers

Antimicrobial resistance (AR) has been exacerbated by the inappropriate use of diagnostics, leading to excessive utilization of empiric antibiotics. The money invested in laboratory infrastructure with 24 h operation has contributed to lower AR rates [88]. The strategy of routine microbiologic surveillance has tended to be replaced by the diagnostic-driven philosophy, aimed at promptly identifying the source of the infection [41]. The insufficient capacity of diagnostic laboratories in LMICs is one of the hurdles in their implementation of AS [30]. Diagnostic stewardship (DS) involves modifying the process of ordering, performing, and reporting diagnostic tests to improve the treatment of infections according to the most updated literature. The proper steps are referred as preanalytic, analytic, and postanalytic interventions [8,89].

There is an emerging role of and opportunities for rapid diagnostics and biomarkers in AS. Point-of-care (or point-of-impact) assays, have become more readily available, and they enable the rapid identification of bacterial pathogens, as fast as in two hours, and affect effective clinical decision making. Innovative methods such as next-generation sequencing provide further hope for earlier diagnosis [20]. Nevertheless, newer diagnostics should not replace culture-based diagnostics using blood and other clinical specimens along with susceptibility testing [20]. The optimal goal is the correct identification of the pathogen in order to focus and de-escalate antibiotic treatment [41]. Unfortunately, in the real world, there are inequities in access to diagnostics between high- and lower- and middle income countries [7].

Diagnostic stewardship comprises the procedures that lead to the choice of the right test for the right patient at the right time. It promotes the judicious use of rapid and novel diagnostic tests that target all pathogens that cause colonization and infection [20]. A patient-oriented and personalized approach is needed where clinical consequences are dependent on timely diagnostics [20]. DS assists in the selection of the proper treatment and the early discontinuation of antibiotics through the interpretation of results. DS also targets the overdiagnosis of infectious diseases, which could lead to unnecessary treatments. It could also be implemented through the rejection of unsuitable samples [88]. In surgery, the diagnostic pathway should involve the correct acquirement of tissue cultures via surgical debridement or peritoneal fluid culture in cases of peritonitis [27,88]. In addition to the rapid and correct identification of pathogens, the biomarkers that depend on the host’s response to infection have offered a new perspective on infectious disease diagnoses in patients undergoing surgery [27].

Newer methods such as MALDI-TOF, PCR, antigen-based tests, NAAT, bacterial DNA enrichment, and amplicon hybridization on microarrays have been developed and successfully decreased the time until pathogen identification; they can also be used to detect resistant traits [8,88,90]. These new methods can provide data more rapidly and increase detection sensitivity. Physicians associate them with positive clinical, logistic, and financial outcomes. They could be especially useful in detecting pathogens that are difficult to grow or those antibiotic-pretreated patients, significantly reducing hospital costs [91,92].

The perfect biomarker for guiding AS in surgery has not yet been defined [93]. Obviously, any infection is far too complex to be reduced to a single cutoff for any biomarker [94]. Procalcitonin (PCT), s along with C-reactive protein (CRP), is the best studied biomarker that guides antimicrobial treatment in this setting. Its use has not been completely elucidated, as surgical patients have comprised a minority of the large studies evaluating the method [11,93,95]. Major stresses such as surgery may elevate procalcitonin levels through the gut translocation of bacterial products [11,96]. In several studies regarding the clinical use of PCT to reduce antibiotic consumption, patients that have undergone major surgical procedures such as cardiac surgery and lung transplantation have not been included [11,96]. When studied in cardiac surgery, PCT was a better biomarker than CRP for identifying infective inflammatory responses [93]. In addition, it was proposed that higher cutoffs for PCT should be used for postsurgical patients [11]. The levels of other sepsis biomarkers such as CRP and IL-6 also increase in the first hours after surgery. The advantage of PCT is that its levels fall rapidly, so any further increase probably correlates with sepsis [45,93]. In patients with sepsis in surgical ICUs, PCT has been found to be a useful biomarker to safely reduce antibiotic consumption in this setting [11,97,98]. CRP, overall, is considered a nonspecific but highly sensitive indicator of an inflammatory response [27].

A multidisciplinary team of experts attempted to define the role of PCT across the surgical pathway and the ability of PCT-guided algorithms to assist AS programs in reducing antibiotic use [39,43]. They concluded that PCT may be useful in guiding the duration and cessation of antibiotics and consequently the exposure to them in patients with acute peritonitis [43,99]. Along with clinical evaluation, especially in the case of progressive or persistent organ failure, the PCT value and its trend can guide the decision for a relaparotomy [43,99]. The optimum length of antibiotic treatment in patients with severe sepsis after abdominal surgery was evaluated in the study by Schroeder et al. Daily PCT measurements were assessed as a guide for the length of antibiotic treatment [100]. The improvement in the clinical signs of infection and a decrease in the PCT levels within three consecutive days contributed to an optimized antibiotic regimen without adverse effects on therapeutic success [100].

In acute diverticulitis, C-reactive protein (CRP) is the most commonly evaluated marker, but it has only shown moderate sensitivity and specificity. Recent data indicate a role of fecal calprotectin or matrix metalloproteinase in detecting diverticular disease, but it has not been clarified whether they are useful for discriminating between complicated and uncomplicated cases [101]. However, PCT could be an interesting parameter in differentiating between the two forms of the disease and deter the unnecessary use of antibiotics [43].

In patients with acute necrotizing pancreatitis, PCT monitoring could aid in the decision to escalate or discontinue antibiotics, but the cutoff levels still need to be determined [43,99]. PCT-guided treatment (with a cut-off of >0.5 ng/mL) proved to be superior to 2-week prophylactic treatment in patients with severe acute pancreatitis [43]. In addition, PCT has been proven to be superior compared with biomarkers like CRP in diagnosing infection in the early phase of acute pancreatitis and in predicting the likelihood of developing infected pancreatic necrosis [43,71]. CRP is part of the inflammatory response in acute pancreatitis; therefore, an upward trend in its levels should not be interpreted as an indicator for antibiotic treatment [43].

There is controversy regarding if PCT exhibits any superiority compared to other biomarkers in diagnosing acute appendicitis [39,102]. However, there is no single biomarker that can differentiate between complicated and uncomplicated disease [39]. In acute cholecystitis, an early high PCT level upon admission had a high sensitivity in predicting major complications [39].

PCT has been superior to CRP in diagnosing early sepsis in patients with trauma and after a neurosurgical procedure [43]. When trauma patients with a poor prognosis were compared to those with a good prognosis, there were significant differences in the levels of PCT, CRP, and IL-6 [43]. As mentioned above, trauma and its consequences, as well as the surgical procedures and the treatment in the intensive care unit, might influence the PCT concentration [43]. However, a persistent increase in the biomarker is a good indicator of developing complications of sepsis [43]. In the review by Parli et al., the significance of PCT was emphasized, but no standard approach to its use in trauma patients was recommended [103]. In the study by Chomba et al., the investigators were able to reduce the duration of antibiotic treatment in trauma when guided by PCT levels; however, the results did not reach statistical significance [104]. Notably, there was an indication that, in trauma patients, the biomarker use may have the potential to be used to reduce both antibiotic usage and costs [104]. The role of biomarkers in SOT recipients has not been fully elucidated. Immunosuppression may alter the PCT results, altering their reliability [74].

### 2.5. The Antibiotic Stewardship Team—New Technologies and Antimicrobial Stewardship

An AS program should involve a multidisciplinary team. The composition of the AS team may differ across countries and health systems [22]. The process is mainly seen as a task for infectious disease (ID) specialists; clinical microbiologists; hospital pharmacists; and infection prevention, hygiene, and environmental medicine specialists. As patients move through the hospital, however, we cannot ignore the role of bedside doctors, nurses, and more importantly the board of directors, who provide the financial resources [8,10,20,21,46,105]. The directors should ascertain that the staffing standards are met based on the list of the actions that the team should implement and that a sustainable funding mechanism is established for dedicated practitioners to be employed [106]. It is very important that the input of a variety of healthcare professionals is promoted, along with the more-active participation of nurses and a better-defined role for pharmacists [22].

The highly interactive, face-to-face approach of discussing diagnosis and treatment is especially important for complex patients and requires the participation of senior team members. Under the threat of antimicrobial resistance and the difficulty of treating infections, it is necessary to cross borders and approach infection management in an integrated manner [20,21]. If we consider that a particular patient may move among different hospitals, infection management affects patients through the entire network [20,21]. The team should be able to implement educational and restrictive measures and have the ability to monitor antibiotic use [19,20,51,106]. They should provide systematic expert advice in certain cases of MDR pathogens or complicated infections and restrict the prescribing of antibiotics [19]. It is very important that infectious disease teams acquire a real-time assessment of the local microbial epidemiology [89]. Infectious disease consultation provides a very effective way to improve the clinical outcome of surgical patients with infectious syndromes. The problem is that most antimicrobials used in surgical departments are not prescribed by infectious disease physicians, and the patients do not have a formal ID consultation despite the fact that it could have an impact if performed at the right moment [107]. In these times, there is a tremendous opportunity for infectious disease physicians to put their knowledge and expertise in action in order to implement proper changes in antimicrobial regimens [1].

The selection of the right antibiotic is pivotal in the treatment of sick patients. To this end, the identification of microbial genes related to drug resistance directly from clinical specimens or from positive blood cultures reduces the time to result and enables the provision of effective treatment promptly [108]. The role of the clinical microbiologist in the multidisciplinary team varies across the health systems. In low- and middle-income countries, the microbiology laboratory is mainly a service department, whereas, in other healthcare models, the medical microbiologist is directly involved in the patient’s care [88]. The clinical microbiologist may also play a major role by improving the diagnostics and report susceptibilities in due time by using rapid methods that can also identify an emerging range of resistance mechanisms [88]. Their interventions also include mapping the local epidemiology, detecting early outbreaks, and defining local empirical antibiotic regimens [88]. The AS team could also assist in the AS process by introducing special antibiograms in order to facilitate empiric treatment and by endorsing “protected” antibiotic formularies [8,109]. ID pharmacists are well-qualified members of the AS team. The Infectious Disease Society of America (IDSA) and the Society for Healthcare Epidemiology of America (SHEA) promote the co-leadership of ID physicians and pharmacists as the best model for effective stewardship [105].

Artificial intelligence (AI) is becoming increasingly important in healthcare because of its ability to turn data and information into insight and knowledge [110,111]. The important advantages of AI include its independence from behavioral and hierarchical constraints and peer influence and its commitment to guidelines [110]. Supervised and unsupervised machine learning (ML) tools can predict antibiotic resistance, help select the appropriate treatment, and guide dosing adaptations and drug–drug interactions, especially in the care of vulnerable populations [109,110,112]. This is achieved through complex processes that facilitate optimal data use, with the end point being the development of prediction tools [110]. These systems depend on the development of a knowledge base and the direct learning of the rules from a massive volume of data, and they can adapt to unknown situations without instructions [109,112]. ML is also invaluable in research because it allows scientists to focus on more complex matters [110]. However, we should always keep in mind that these models lack generalizability and should be adapted to different healthcare settings due to differences in local susceptibility patterns [110]. At a more advanced level, ML algorithms can combine patients’ microbiome profiles with laboratory data in order to individualize antibiotic resistance control [110,111].

Several clinical decision-support systems (CDSSs) could greatly improve antimicrobial stewardship and guide treatment [109,110,112]. These systems are traditionally based on human expertise but, when they are ML-assisted, they can learn automatically and improve knowledge obtained from data and interpret unknown situations [110]. Attention should be paid to the variables that are integrated in ML-CDSSs and to the need for high-quality clinical databases [110]. Additionally, the integration of antibiograms into CDSSs provides the opportunity to administer appropriate antibiotic therapy and help with personalized pharmacokinetics for patients [109]. The end point of the use of these advanced modalities should be the early detection of AMR and the administration of the ideal treatment to the patient [110,111].

Education on ML is another significant facet of the development of new technologies in hospitals. For physicians with limited knowledge, automated ML (autoML) could provide a valuable tool. However, the default configurations are not suitable for all applications, and judicious usage is associated with a fundamental understanding of the underlying algorithms [110]. ML models can suffer from biases; thus, the deployment of such technologies can be a complex issue [110].

## 3. Materials and Methods

The aim of this review was to present a comprehensive picture on the challenges facing AS programs in surgical departments and how such programs may become more efficient. Special mention was made of the very important matter of treatment duration and the role of the microbiology laboratory. To address the above, the authors independently searched “PubMed” using the terms “surgical site infections”, “surgical antibiotic prophylaxis”, “antimicrobial resistance”, “antimicrobial stewardship in surgery”, “Diagnostic stewardship” and “biomarkers”. The authors focused on the literature in the past decade; however, important articles beyond the 10-year limit were also included. In total, 112 articles were included in this review.

## 4. Conclusions

In this article, we attempted to address the all-important issue of the implementation of AS principles in surgical departments. The aim was not to include all the articles available on the matter. This is a significant limitation of the article; however, we believe that we adequately addressed this important issue, and significant information is included. In the ever-increasing era of antibiotic resistance, the threats of inefficient SAP and the subsequent SSIs are imminent. There are several reasons that lead to the improper use of antibiotics in surgical wards (summary in Figure 1). AS teams should focus on mitigating the effects of these factors. Further research is needed on the role of the decolonization of resistant bacteria before surgical operation. There is also a need to establish the use of broader-spectrum or targeted alternative agents in patients colonized with resistant bacteria. A targeted approach is recommended. The close collaboration between AS and surgical teams, along with an enforced diagnostic laboratory, is pivotal for the optimization of the care of patients undergoing surgery.

Key Points:HAIs affect 5-15% of hospitalized patients, leading to increased vulnerability to MDROsAS objectives include empirical therapy suggestion, therapy de-escalation, IV to oral switch, and discontinuation of empirical treatment when no evidence of infectionAS aims for optimal clinical outcomes and reduction in antimicrobial resistanceEvidence-based guidelines and education are essential for reducing infections and AMRAS interventions should be tailored to local conditions and implemented through behavior-changing techniquesAS in surgical wards should focus on SSI prevention and SAPSSIs occur in 1-3% of inpatient surgeries, contributing to patient injury, mortality, and increased healthcare costsProper antibiotic use in surgery faces challenges such as unclear indications and lack of adherenceEducation plays a vital role in AS programs, including continuous training on AMRDS involves choosing the right test for the right patient at the right time, utilizing rapid diagnostics and biomarkers for prompt pathogen identification

## Figures and Tables

**Figure 1 antibiotics-13-00329-f001:**
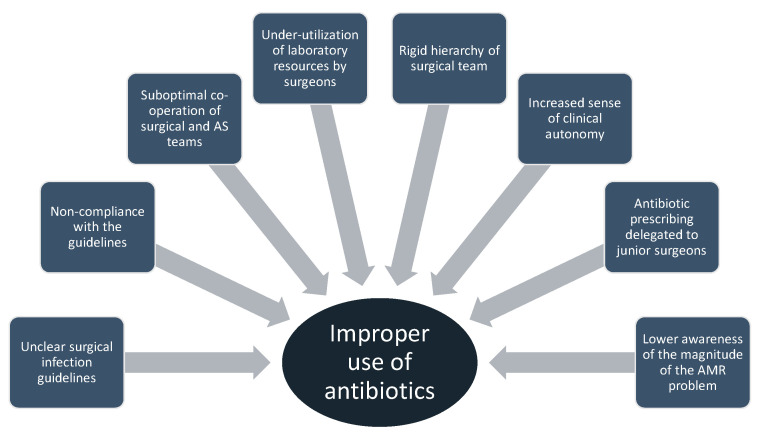
Reasons that lead to improper use of antibiotics in surgical wards.

**Table 1 antibiotics-13-00329-t001:** The most common SSI-causing pathogens in different time periods and the surgeries in which they are prevalent (modified from Young et al. and Bassetti et al. [36,37]).

Pathogen	Variation over Time			Type of Surgery Where Prevalent
	1990–1996	2007	2010	
*S. aureus*	20%	30%	30.4%	Prostheses, implants
				Cardiac
				Neurosurgery
				Ophthalmic
				Orthopedic
				Vascular
*Coagulase negative* *staphylococci*	14%	13.7%	11.7%	Prostheses, implants
				Cardiac
				Neurosurgery
				Ophthalmic
				Orthopedic
				Vascular
*Enterococcus* spp.	12%	11.2%	11.6%	Colorectal
				Obstetrics and gynecology
Gram-negative bacilli				Appendectomy
*E. coli*	8%	9.6%	9.4%	Biliary tract
*P. aeruginosa*	8%	5.6%	5.5%	Colorectal
*Enterobacter* spp.	7%	4.2%	4%	Gastroduodenal
*Klebsiella* spp.	3%	4%	4%	Obstetrics and gynecology

**Table 2 antibiotics-13-00329-t002:** Changes in the resistance patterns of SSI-associated pathogens (percentages) in two different time periods (modified from Sievert et al. [38]); MDR: multidrug-resistant; R: resistant.

	2007–2008	2009–2010
Methicillin R *S. aureus*	48.0	43.7
Vancomycin R *Enterococcus*		
*E. faecium*	65.2	65.3
*E. faecalis*		
*Klebsiella* spp.		
Cephalosporin R	19.4	13.2
Carbapenem R	9.6	7.9
MDR	10.9	6.8
*E. coli*		
Cephalosporin R	9.1	10.9
Fluoroquinolone R	27.2	28.3
Carbapenem R	1.5	2.0
MDR	1.1	1.6
*Enterobacter* spp.		
Cephalosporin R	30.6	27.7
Carbapenem R	2.8	2.4
MDR	1.5	1.7
*Pseudomonas aeruginosa*		
Aminoglycoside R	4.4	6.0
Cephalosporin R	13.6	11.2
Fluoroquinolone R	15.8	16.9
Carbapenem R	11.2	11.6
Piperacillin/Tazobactam R	6.8	6.8
MDR	4.9	5.3
*Acinetobacter baumannii*		
Carbapenem R	38.6	37.3
MDR	49.3	43.9

**Table 3 antibiotics-13-00329-t003:** Compliance rates for indication, timing, duration, and antibiotic choice for different kinds of surgical procedures as reviewed in the study by Agodi et al. [63].

Type of Surgery	Indication	Timing	Duration	Antibiotic Choice	Overall
Orthopedic	99.4%	73%	70.2%	57.7%	43.6%
Abdominal surgery including herniorrhaphy					12.4%
Cardiac		97.4%	66.1%		
Clean or clean-contaminated	18.6%	30.3%	26.7%	30.8%	9.4%
Caesarean section					59.5–100%
Vascular	95%		87%	98%	80%
Emergency trauma laparotomy					
Various	44.8–95%	53.4–100%	59.4–88.7%	25.5–98%	6.9–100%

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
