# Peer review of "Antibiotic Stewardship in Surgical Departments"

_antibiotics, 2024, doi:10.3390/antibiotics13040329_

Round 1
Reviewer 1 Report (New Reviewer)
Comments and Suggestions for Authors
The paper by Kourbeti, Kamiliou, and Samarkos, "Antibiotic Stewardship in Surgical Departments," presents a comprehensive review of the role of antimicrobial stewardship (AS) in surgical wards, focusing on surgical site infections, surgical antibiotic prophylaxis, and the utilization of diagnostic stewardship and biomarkers. It underscores the criticality of antimicrobial resistance (AMR) and the importance of AS in mitigating this global health threat. While the paper is informative, certain areas could benefit from improvement.
1. Broader Global Perspective: The paper tends to focus primarily on high-income countries, with less emphasis on the unique challenges and solutions pertinent to low and middle-income countries (LMICs). A more inclusive global view, addressing the variations in resources, infrastructure, and healthcare systems, would provide a more rounded perspective.
2. Clarity and Conciseness: The paper is densely packed with information, which while comprehensive, sometimes affects readability and clarity. Streamlining content, using more visual aids (charts, graphs, tables) for data representation, and summarizing key points could enhance understanding and retention of the material.
3. In-Depth Analysis of New Technologies: While the paper touches upon the role of artificial intelligence (AI) and machine learning in AS, it lacks depth in exploring how these technologies could specifically revolutionize AS in surgical departments. Detailed case studies or recent research findings could provide practical insights into the implementation and effectiveness of these technologies.
4. Interdisciplinary Approach: The paper emphasizes the role of AS teams but could expand on the interdisciplinary nature of tackling AMR, particularly the roles of nursing staff, surgical teams, and hospital administration in implementing AS programs. Highlighting successful case studies of interdisciplinary collaboration could offer valuable learning points.
5. Policy and Implementation Strategies: While the paper thoroughly discusses the importance of AS, it could benefit from a more detailed exploration of policy-making, implementation strategies, and the challenges therein. Insights into how different hospitals or health systems have successfully incorporated AS programs would be beneficial, especially regarding policy advocacy and stakeholder engagement.
Conclusion
"Antibiotic Stewardship in Surgical Departments" is a valuable resource in the field of healthcare, particularly in the context of AMR. However, addressing the aforementioned points could greatly enhance its impact and utility for a broader range of healthcare professionals and policymakers. The authors have laid down a solid foundation, and with these improvements, the paper has the potential to significantly contribute to the global discourse on antimicrobial stewardship.
Comments on the Quality of English LanguageModerate editing of English language required
Author Response
Dear Reviewer,
we wish to thank you for the very constructive comments,. We are submitting a point bt point response to them
Sincerely
the authors

Reviewer 2 Report (New Reviewer)
Comments and Suggestions for Authors
The article concerns a review of the scientific literature relating to a topic of great medical interest: antimicrobial resistance. The review is clear and comprehensive, but it needs to be expanded further both in terms of methods and conclusions. It also doesn’t have a well structured manner. The completeness of the review topic is not completely covered. Bibliographic citations may need to be reported more accurately and carefully in order to reduce the possibility of inconsistencies in the data reported within the article.
The following is also suggested:
Introduction: considering what the purpose of the article is, i would describe more in the introduction what both programmes and AS teams are and what their objectives are.
Methods: expand materials and methods by making the time window of the study explicit and better describing the procedures used. Increase both the number of search engines used and the keywords. Explain the inclusion and exclusion criteria for articles in the review.
Results and discussions: it would be interesting to list the number of articles
consulted.
It is advisable to further investigate the purpose of the study listed in the introduction and make them clearer (special challenges that the AS teams face in the surgical departments, the updated data on the rational antibiotic use and the shorter treatment durations for surgical infections, the special roles of each member of the AS team and the importance of the diagnostic laboratory).
Line 53: the quotation appears to be inaccurate. In fact, the quoted article (reference16) refers to another study in which it is stated that the estimated cost reported relates to US healthcare expenditure for five of the largest HAIs. We therefore recommend reading the following article: Zimlichman E, Henderson D, Tamir O, et al. Health Care–Associated Infections: A Meta-analysis of Costs and Financial Impact on the US Health Care System. JAMA Intern Med. 2013;173(22):2039–2046. doi:10.1001/jamainternmed.2013.9763
Line 97: it is advisable to write the acronym SSI in full as it is the first time you buy into the article.
Line 128: The data in Table 1 and Table 2 need to be updated. Improve the layout of table 1. The association between percentages and gram-negative bacilli is not understandable. Improve the layout of table 2 for instance by inserting column headings in such a way as to make them easier to understand.
Line 167: it may be necessary to elaborate and report the figure more clearly (40%). In fact, a study on the use of antibiotics in Scottish hospitals is cited here.
Line 265: report the data more accurately in Table 3. For example, with regard to caesarean section, the 100 per cent figure in the Agodi et al study referred to concerns the timing of PAP. You should also specify what you are referring to with “overall” (indicators used are not always the same). It is also advisable to look more closely at the indicators referred to.
Line 343: it might be interesting to elaborate on this statement “including patients who are immunosuppressed”.
Line 372: in relation to "DS" write in full what an acronym means the first time it is used in the article.
Line 415: Considering the prevalence in the population, it might be interesting to investigate antibiotic resistance also in lung diseases and possible solutions to this, which is why it might be interesting to explore the topic further by reading and quoting the following article: Troiano G, Messina G, Nante N. Bacterial lysates (OM- 85 BV): a cost-effective proposal in order to contrast antibiotic resistance. J Prev Med Hyg. 2021 Jul 30;62(2):E564-E573. doi: 10.15167/2421-4248/jpmh2021.62.2.1734. Erratum in: J Prev Med Hyg. 2021 Sep 15;62(3):E790-E792. PMID: 34604601;PMCID: PMC8451348.
Conclusion: it is advisable to respond more clearly to the objectives of the study listed in the introduction. It might also be interesting to expose what the limitations of the study are.
Comments on the Quality of English LanguageEnglish language is appropriate and understandable.
Author Response
We would like to thank the reviewer for his comments
We are submitting a point by point response to them
Sincerely
the authors

Round 2
Reviewer 1 Report (New Reviewer)
Comments and Suggestions for Authors
The authors have excellently revised the manuscript. Congratulations! As the manuscript is of extremely high importance for clinicians and scientist worldwide, it should be published as soon as possible!
Author Response
We would like to thank the reviewer for their kind words. Their remarks have been invaluable for the improvement of the paper
Reviewer 2 Report (New Reviewer)
Comments and Suggestions for Authors
Dear authors Looking at the manuscript, I note that you have only
accepted some of the changes that were previously recommended to you.
In order to improve the article in addition to those I would like to suggest
further changes.
Introduction:
A definition of what an as-team is and what its possible members are
might make the objective of the study more understandable already from
reading the introduction to the study.
Methods:
It might be interesting to know the duration and when the study took place.
Results and conclusion:
Table 3: We think you should more accurately report the information in
this table by referring to the study mentioned. (Orthopaedic surgery only
refers to hip or knee replacement, “overall” indicators used in the in the
various studies reported are not always the same).
Comments on the Quality of English LanguageEnglish is not very fluent, therefore needs to be improved.
Round 3
Reviewer 2 Report (New Reviewer)
Comments and Suggestions for Authors
The article concerns a review of the scientific literature relating to a topic of great medical interest: antimicrobial resistance. The review needs to be expanded further both in terms of methods and conclusions. The following is also suggested:
Methods: Expand materials and methods by making the time window of the study explicit and better describing the procedures used. Increase both the number of search engines used and the keywords. Explain the inclusion and exclusion criteria for articles in the review.
In particular Selection of methods:
Inclusion and exclusion criteria: Define the criteria used to select the methods to be included in the review, specifying the characteristics of the studies (e.g. type of study, population, intervention) and the sources of information (e.g. databases, scientific publications).
Search strategy: Describe in detail the procedure used to identify the relevant methods, including the keywords, databases and other search tools used.
In particular Evaluation of methods:
Assessment tools: Present the tools used to assess the quality and reliability of the methods, such as checklists or evaluation grids.
Evaluation criteria: Specify the criteria used for the evaluation, such as the internal validity, reliability, transferability and applicability of the methods.
In particular Data analysis and synthesis:
Data extraction: Describe the process of extracting data from the selected methods, specifying the information collected (e.g. method characteristics, results, conclusions).
Data analysis: Present the analysis of the extracted data, using appropriate statistical or qualitative techniques.
Data synthesis: Synthesize the results of the analysis and present a critical overview of the methods examined, highlighting their strengths and weaknesses.
Results and discussions: it would be interesting to list the number of articles consulted.
.
Line 284: report the data more accurately in Table 3. For example, with regard to caesarean section, the 100 per cent figure in the Agodi et al study referred to concerns the timing of PAP. You should also specify what you are referring to with “overall” (indicators used are not always the same). It is also advisable to look more closely at
the indicators referred to.
Line 433: Considering the prevalence in the population, it might be interesting to investigate antibiotic resistance also in lung diseases and possible solutions to this, which is why it might be interesting to explore the topic further by reading and quoting the following article: Troiano G, Messina G, Nante N. Bacterial lysates (OM-85 BV): a cost-effective proposal in order to contrast antibiotic resistance. J Prev Med
Hyg. 2021 Jul 30;62(2):E564-E573. doi: 10.15167/2421-4248/jpmh2021.62.2.1734.
Erratum in: J Prev Med Hyg. 2021 Sep 15;62(3):E790-E792. PMID: 34604601;
PMCID: PMC8451348.
Conclusion: It might be interesting to expose what the limitations of the study are.
Line 979 “Figure 1. Reasons that lead to improper use of antibiotics in the surgicalwards”: the two figures are identical.
The English used is readable
Author Response
Please see the attachment

This manuscript is a resubmission of an earlier submission. The following is a list of the peer review reports and author responses from that submission.
Round 1
Reviewer 1 Report
Comments and Suggestions for Authors
The manuscript consists of total 15 pages, including the list of total 103 literature references. The review article provides insight into the problem of antibiotic resistance spreading prevention-related activities in surgical wards environment. As such, the article is potentially interesting to the Journal's Readers and fits into the scope of works published in the Journal. The English language used in the text is of satisfactory quality. The title of the manuscript is concise and informative enough, relevant to the contents of the article. All the abbreviations (AMR, AS etc) shall be explained while used for the first time, in the Abstract and in the main text of the article.
The Abstract is not structured but it encompasses the key aspects risen in the main text of the article.
The Introduction provides enough background information justifying undertaking the study.
The Results and Discussion section are merged, which is acceptable but not advisable for a scientific publication. This section is detailed enough and divided logically into thematic sub-sections, which aids rapid orientation in the text.
The Material and methods section is detailed enough.
The Conclusions section summarizes the key findings; abbreviations used here shall be explained again for the benefit of the Readers who start to read the article from this section in order to decide whether to read it in full.
line 22 - "...becomes one of the leading causes..." would be more appropriate
line 32 - "...decide to treat only the..." would be more appropriate
line 33 - "...suffice as well as determine by the microbiology laboratory tests the least wide-spectrum antibiotic suitable in the given case and switch to it in the therapy as early as possible." would be appropriate
line 50 - "team" is meant?
line 121 - NNIS - in which country?
line 174 "...C. difficile infections..." - this statement may be questionable as proper perioperative antibiotic prophylaxis is typically short enough to avoid the selection of C. difficile in bowels, as the Authors state in the line 245
Summing up, I liked reading this work a lot as it presents the current state of art in a disciplined and clear way.